# A Method for Assessing Dogs in a Test Evaluating Dogs’ Suitability for Animal-Assisted Education

**DOI:** 10.3390/ani14081149

**Published:** 2024-04-10

**Authors:** Weronika Stempiń, Janusz Strychalski

**Affiliations:** Department of Fur-Bearing Animal Breeding and Game Management, University of Warmia and Mazury, Oczapowskiego 5, 10-719 Olsztyn, Poland; weronika.stempin@student.uwm.edu.pl

**Keywords:** behavior evaluation, behavior test, dog behavior, therapy dogs

## Abstract

**Simple Summary:**

Objective behavioral rating (BR) is one of the methods for assessing dog behavior. In this approach, dogs are rated for behavior based on the observations made during a behavioral test. However, the degree of agreement among independent raters can differ. These variations could be attributed to different or unclear definitions of behavioral patterns. Thus, the aim of this study was to propose a new approach for assessing dogs’ suitability for animal-assisted education (AAE) with the use of the BR method and the definitions of dog behaviors proposed by artificial intelligence (AI). The analysis relied on video footage of dogs that participated in tests assessing the animals’ suitability for AAE. The results indicate that the BR method based on AI definitions produced satisfactory inter-rater reliability in a test evaluating dog behavior. A novel finding of this study is that the BR method can be used in dog assessments based on temperament traits and momentary emotional states and that dog behaviors can be evaluated based on the definitions proposed by AI. Thus, we conclude that the proposed approach gives promising outcomes and can be used to develop new tests for assessing dogs’ suitability for AAE.

**Abstract:**

In objective behavioral rating (BR), dogs are rated for behavior based on the observations made during a behavioral test. However, behavioral definitions can differ between raters, and the definitions proposed by artificial intelligence (AI) can help standardize the applied measures. The aim of this study was to propose a new approach for assessing dogs’ suitability for animal-assisted education (AAE) with the use of the BR method and the definitions of dog behaviors proposed by AI. The analysis relied on video footage of 25 dogs that participated in tests assessing the animals’ suitability for AAE. The dogs were rated by five independent observers. Inter-rater reliability was satisfactory in 7 out of 9 subtests (0.715–0.856) and low in the remaining 2 subtests (0.667 in Subtest 2 and 0.617 in Subtest 5). The results indicate that the BR method based on AI definitions produced satisfactory inter-rater reliability in a test evaluating dog behavior. A comparison of subtest scores in the BR method and the subjective rating method did not reveal significant differences. Thus, it can be concluded that the proposed approach gives promising outcomes and can be used to develop new tests for assessing dogs’ suitability for AAE and, perhaps, other types of work.

## 1. Introduction

According to the definition proposed by Pet Partners (former Delta Society), animal-assisted interventions (AAIs) are therapeutic interventions in healthcare, education, and human services that are based on emotional, psychological, and physical interactions between humans, animals, and the environment, and are undertaken to improve health and wellness [1]. Animal-assisted interventions include animal-assisted therapy (AAT), animal-assisted education (AAE), and animal-assisted activities (AAAs). A recent study proposed that the term AAI should be replaced with animal-assisted service (AAS) as an umbrella term and animal-assisted treatment (AATx), animal-assisted education (AAE), and animal-assisted support programs (AASP) as subcategories [2]. Animal-assisted therapy generally involves a client, a therapist, an animal (typically a dog), and an animal handler, where the therapist and the animal handler can be the same person [3]. Therapies based on interactions with animals can involve various types of activities, including physical activities such as feeding and grooming the dog; cognitive activities such as puzzles; and social activities such as introducing, greeting, praising, thanking, and talking to each participant and the dog. Therapeutic activities such as singing or teaching a dog to find treats can also elicit positive emotions [4].

Most importantly, dogs participating in AAIs should have a friendly attitude toward the world and should be patient and willing to cooperate with humans. Dogs that are aggressive, territorial, and excessively anxious are not suited for AAIs. Similarly to Germany and Norway, Poland does not have an official examination system for therapy dogs because these countries have not developed legal regulations concerning canine-assisted therapy [5]. As a result, dogs are trained, evaluated, and selected for AAI programs mainly by non-profit organizations. The Personality Assessment of Domestic Animals (PADA) composed of 18 exercises is a good example of a test that could be applied universally [6]. Some organizations subject therapy dogs to preliminary tests to ensure that they are fit for the job. These tests aim to determine whether the dog copes well with stress and whether it requires professional training. The purpose of preliminary tests is to identify dogs that are unsuitable for AAIs and to eliminate these animals from further training [7].

There are many methods for measuring dog behavior. These methods can be divided into two general categories: (1) methods that measure the quantity and frequency of selected behaviors (behavior codings), such as tail wagging, and (2) methods that assess dog behavior (behavior ratings). Dog behavior can be evaluated with the use of objective behavioral rating (BR) or subjective rating (SR). In the former approach, dogs are rated for behavior based on the observations made during a behavioral test, such as the animal’s reaction to a child-like doll [8], whereas in the latter approach, dogs are rated for specific traits, such as confidence [9]. In the BR method, the dog is usually placed in various standardized situations, such as a stranger touching the dog. Operational definitions of possible dog behaviors are prepared. Human raters then match the observed behavior to these definitions. A behavioral test has to be validated to assess its reliability and the reliability of the results. The validity of a test is commonly evaluated based on inter-rater agreement, where the greater the agreement between raters, the higher the reliability of the test [10]. Dog behaviors can also be evaluated with the Qualitative Behavioral Assessment (QBA) approach, where the raters focus on how the animal behaves and interacts rather than what the animal does [11,12].

Research on animal behavior has important implications for many fields of biology and biomedical sciences, which is why artificial intelligence (AI) solutions are increasingly often used in this area [13]. The number of predefined behaviors exhibited by animals is counted by dedicated software. In most contemporary programs, specific behaviors are identified by tracking “high-level” properties, such as the location of body parts in space (characteristic poses) and the speed with which different body parts move [14,15]. Some algorithms are able to learn how to identify and count selected behavioral patterns. These programs can improve their ability to correctly recognize and quantify specific behaviors [16]. However, it should be stressed that AI tools are useful for measuring the quantity and frequency of selected behaviors (behavior codings), whereas dog behavior is still evaluated by humans in multi-stage behavioral tests. Therefore, regardless of the applied rating method (objective or subjective), the degree of agreement among independent raters can differ [9]. These variations could be attributed to different or unclear definitions of behavioral patterns. In the near future, humans could become inspired by the definitions proposed by AI. This is one of the ways in which AI can help assess and predict dog behaviors and prevent problems before they occur.

The aim of this study was to propose a new approach to assessing dogs’ suitability for animal-assisted education (AAE) by the BR method based on the definitions of dog behaviors proposed by AI, which was validated by analyzing inter-rater reliability. Assessments of dog behavior by the SR method, where dogs were evaluated for sociability, touch tolerance, and overall suitability for AAE, were used as the control.

## 2. Materials and Methods

The analysis relied on video footage of 25 dogs and their owners who participated in tests assessing the animals’ suitability for AAE. The animals were purebred and mixed-breed male and female dogs of different ages and origins. The dogs’ suitability for AAE was assessed by an experienced trainer (certified dog therapist and COAPE behaviorist), referred to as the test leader (TL). Fifteen assistants also took part in the test. The test was performed in a closed room measuring 10 × 15 m, with a ceramic tile floor and large windows. The test had been described previously [7], and it consisted of nine subtests (stages):The owner walks a leashed dog diagonally across the room from the entrance door. The assistants stand still at various points in the room. This subtest provides information about a dog’s attitude towards a new environment and strangers standing in a neutral position.The owner walks a leashed dog back to the entrance door. The assistants walk around the room. This subtest provides information about a dog’s attitude towards strangers moving neutrally in various directions.The owner walks a leashed dog diagonally across the room from the door. The assistants walk around the room, and approximately every second, one of the assistants hits the floor loudly with a stick. This subtest provides information about a dog’s responses to loud sounds in the presence of strangers moving neutrally in various directions.The dog is unleashed, the owner remains neutral, and the assistants stand still at various points in the room. This subtest provides information about a dog’s general tendency to approach novel objects and willingness to take risks.The owner calls the dog and puts it on the leash. The assistants stand still at various points in the room. This subtest provides information about a dog’s willingness to stay in contact with the owner.The owner gives the dog a “sit down” command. The assistants stand still at various points in the room. This subtest provides information about a dog’s ability to work in a novel environment.The TL crouches and touches the dog’s sides, back, fore and hind paws, and head. This subtest provides information about a dog’s tolerance to being touched in various parts of the body, including the paws.The TL stands approximately 2 m in front of the dog, squeezes a squeaking toy, and drops it before the dog. The assistants stand still at various points in the room. The purpose of this subtest is to determine whether a dog would have an interest in the toy after a possibly stressful experience. It may also provide information about playfulness and the dog’s favorite way of playing.The TL throws a hard object which lands approximately 1 m behind the dog with a loud noise; the assistants stand still at various points in the room. This subtest assesses a dog’s sensitivity to unexpected noise and coping strategies in such a situation.

The dogs’ behaviors were standardized with the use of ChatGPT, an AI tool [17]. Questions concerning dog behaviors were input into ChatGPT, and the answers (abridged by the authors who are COAPE-certified animal behaviorists) are presented in Table 1.

Based on video footage, dog behavior was analyzed by five independent raters: a COAPE-certified animal behaviorist (B), an owner of a certified rescue dog (R), an ordinary dog owner (O), a veterinarian running a private practice (V), and a certified dog trainer who owns a certified therapy dog (T). The raters used Table 1 (excluding Column 1 containing questions) in the evaluation process. Dog behaviors were assessed in each of the nine subtests. The raters also evaluated dogs for sociability, touch tolerance, and overall suitability for AAE on a scale of 1–5 points without relying on the answers provided by ChatGPT.

Inter-rater reliability was evaluated with the use of Cronbach’s alpha statistic. It was assumed that the observers were in agreement when Cronbach’s alpha reached ≥0.700. Distributions of inter-rater reliabilities in subtests and assessments of sociability, touch tolerance, and overall suitability for AAE were compared by the Mann–Whitney U test (with a continuity correction) for independent means. All calculations were performed in R software v. 4.3.2 [18].

## 3. Results

As shown in Figure 1, the overall inter-rater reliability exceeded 0.700 in 7 out of 9 subtests. Internal consistency was lower only in the second subtest (in which the owner walked a leashed dog back to the entrance door) and the fifth subtest (in which the owner called the dog and put it on a leash) at 0.667 and 0.617, respectively. Satisfactory inter-rater reliability was obtained in assessments of sociability (0.701), touch tolerance (0.790), and overall suitability for AAE (0.799). The mean inter-rater reliability was determined at 0.755 for all 9 subtests and at 0.763 for the 3 attributes indicated above.

In Subtests 1–3 (in which the owner walked a leashed dog across the room), inter-rater reliability exceeded the ≥0.700 threshold in only 3 cases (Table 2). A satisfactory result was noted only once in Subtest 1, twice in Subtest 3, and zero times in Subtest 2.

Inter-rater reliability was higher in Subtests 4–6 (Table 3). Satisfactory levels of inter-rater agreement were observed 3 times in Subtest 4 (in which the dog was unleashed) and only once in Subtest 5 (in which the owner called the dog and put it on the leash) (0.827, R vs. O). In Subtest 6 (in which the owner gave a “sit” command), inter-rater reliability exceeded 0.700 on 4 occasions.

Inter-rater agreement was similar in Subtests 7–9 (Table 4). Cronbach’s alpha exceeded the ≥0.700 threshold twice in Subtest 7 (in which the TL crouched and touched the dog’s sides) and 4 times in Subtest 8 (in which the TL held a squeaking toy in front of the dog). High levels of inter-rater agreement were noted twice in the last subtest (in which a hard object landed behind the dog with a loud noise).

In the test where dogs were evaluated for sociability, touch tolerance, and overall suitability for AAE, similar levels of inter-rater reliability were noted in two cases when assessing dogs’ sociability and touch tolerance, and in three cases when assessing the dogs’ overall suitability for AAE (Table 5).

The median values of inter-rater reliability were determined at 0.548 (mean = 0.522) in Subtests 1–9 (sample 1) and at 0.567 (mean = 0.546) in assessments of sociability, touch tolerance, and overall suitability for AAE (sample 2) (Figure 2). No significant differences were found between samples (W = 1290.5, *p* = 0.721).

## 4. Discussion

Behavioral tests have been applied for many years to facilitate the selection of service dogs for different types of work and for breeding purposes [19,20,21]. The present study proposes a new approach to assessing dogs’ suitability for AAE by the BR method based on the definitions of dog behaviors proposed by AI. Assessments of dog behavior by the SR method, where dogs were evaluated for two traits that are essential for therapy work, and for overall suitability for AAE, were used as the control. To the best of the authors’ knowledge, this is the first study to evaluate dogs with the use of the BR method, where the definitions of dog behaviors were provided by AI. Moreover, the operational definitions for each subtest were formulated based on adjectives describing the dogs’ temperament and momentary emotional states (aggressive, anxious, grounded, excited, and reactive) (Table 1). This approach was used to determine whether the definitions proposed by AI can be applied to research on dog behavior. The dogs were rated by five independent observers (a behaviorist, an owner of a rescue dog, an ordinary dog owner, a veterinarian, and a dog trainer who owns a therapy dog) based on the definitions formulated by AI, and inter-rater reliability was satisfactory in 7 out of 9 subtests (0.715–0.856) and low in the remaining 2 subtests (0.667 in Subtest 2 and 0.617 in Subtest 5). Cronbach’s alpha also exceeded the 0.700 threshold in assessments of sociability (0.701), touch tolerance (0.790), and overall suitability for AAE (0.799), where the definitions given by AI were not used. However, each rater had different experiences with dogs, which could explain why the subtest scores and the overall test scores overlapped only partially. This observation is consistent with the general belief that the aggregate scores of multiple observers are reliable and independent of individual observers’ experiences and personality traits [22]. The convergence between expert and non-expert ratings in behavioral assessments of dogs was previously demonstrated by Fratkin et al. [23], although some items were rated with lower reliability and validity by non-experts. According to Kujala et al. [24], rating differences could be attributed to the fact that experts tend to focus on the dog’s posture, whereas non-experts pay most attention to the dog’s head.

Behavioral ratings are less commonly applied than SR to evaluate dog behavior. One of the reasons for the above is that the SR method has a higher capacity for generalization, which could improve its predictive validity [25]. These methods represent different approaches to assessing a dog’s suitability for complex tasks. A dog’s performance will always be evaluated subjectively, even if its behavior or behavioral traits are well defined [26]. Wilsson and Sinn [9] designed a test (composed of 12 subtests) to select working dogs for the Swedish Armed Forces (SAF). In each subtest, potential dog responses to the presented stimuli were standardized using five operational definitions, which is similar to the approach used in the present study. The cited authors also used both BR and SR with the aim of comparing these methods. They concluded that both methods have equal predictive validity in tests assessing dogs’ suitability for military work. Similar conclusions can be drawn based on the results of this study, despite the fact that dogs were evaluated for their suitability for CAE (Figure 2). A different implementation of the BR method was presented by Dowling-Guyer et al. [27]. In the cited study, 38 standardized behaviors were rated during a test composed of 19 subtests.

As part of the standard procedure adopted by many Polish organizations that prepare dogs for AAIs, therapy dogs (and their handlers) have to attend a dedicated training course and pass the final exam to obtain a therapy dog certificate [5]. During the exam, dogs are evaluated for obedience and responses to stimuli. In Poland, the certificate has to be renewed each year. There are no official guidelines or research findings to indicate that tests assessing potential therapy dogs, such as the test proposed in this study, should be repeated. It appears that some qualities that are important in canine-assisted therapy, namely sociability, curiosity/fearlessness, aggressiveness, and boldness, are consistent over time [28]. However, aggressiveness should be treated with caution. This trait may be sensitive to novelty; therefore, a decrease in aggressiveness may be caused by habituation to repeated exposure to the test stimuli [29]. Such observations were made in hens [30], cattle [31], horses [32], and pigs [33]. Repeated tests in dogs produced similar results [28]. However, every dog can behave aggressively in some circumstances, especially in situations that evoke fear. Therefore, not all aggressive behaviors imply that a dog is generally aggressive and should be excluded from AAI.

A novel finding of this study is that the BR method can be used in dog assessments based on temperament traits and momentary emotional states (the reverse approach was used in previous studies) and that dog behaviors can be evaluated based on the definitions proposed by AI. The present study has certain limitations which, however, may pave the way for future research. The main limitation was the use of AI as the source of the definitions. ChatGPT and other AI systems may propose operational definitions of behaviors that are not fully adequate because they are based on online content. The Internet may be a good source of information, but expert knowledge is required to differentiate between information that is true and important and information that is false or not highly relevant. In addition, AI systems keep on developing, and they may provide different answers to a question each time. Therefore, AI should be used with great caution. It should be emphasized that regardless of AI’s assistance in proposing definitions of canine behavior, raters’ assessment will always be subjective. In addition, the study demonstrated that the BR method offers only a simplified set of potential dog behaviors. This set should be expanded to provide the raters with a larger number of options that better fit the observed behaviors, such as the range of behaviors exhibited by anxious dogs. It should also be noted that dogs communicate using complex signals that can have several meanings and undertones. Even subtle changes in facial expression, body posture, and tone of voice could be essential for interpreting dog behavior. Dog behavior can also be ambiguous and ambivalent [34,35]. Therefore, in future research, the raters should have the option of selecting several traits from the provided set of behavioral variables. The test could also be repeated after a specified period of time.

## 5. Conclusions

The results of this study indicate that the BR method based on AI definitions produced satisfactory inter-rater reliability in a test evaluating dog behavior. A comparison of subtest scores in the BR method and the SR method did not reveal significant differences. Thus, it can be concluded that the proposed approach gives promising outcomes and can be used to develop new tests for assessing dogs’ suitability for AAE and, perhaps, other types of work.

## Figures and Tables

**Figure 1 animals-14-01149-f001:**
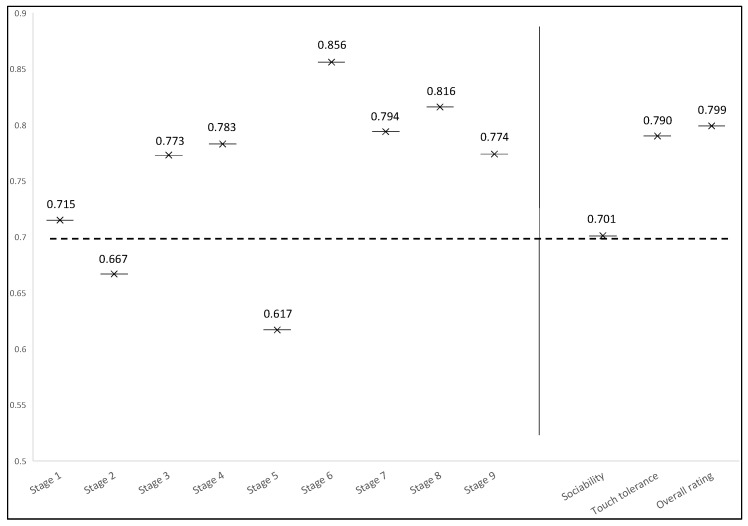
Overall inter-rater reliability.

**Figure 2 animals-14-01149-f002:**
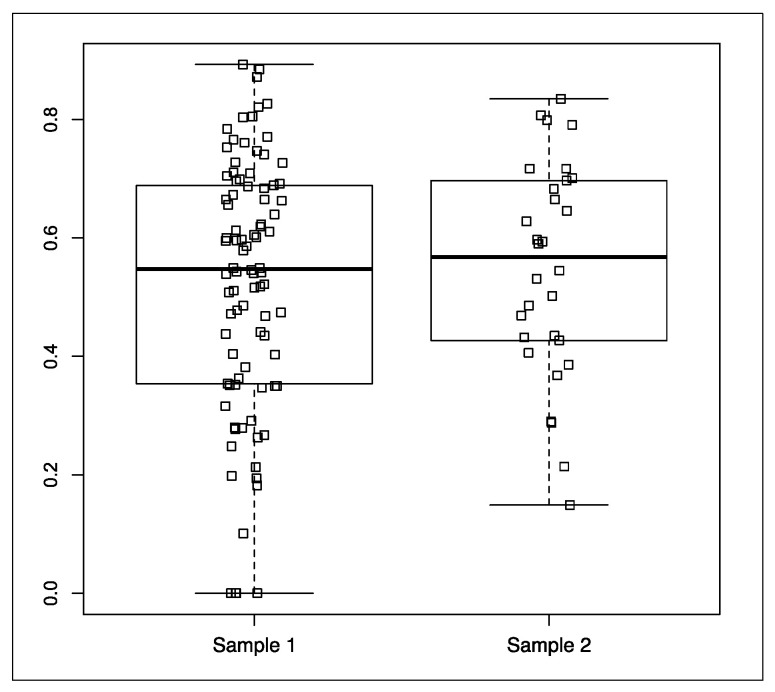
Distribution of inter-rater reliability in Subtests 1–9 (Sample 1) and in assessments of sociability, touch tolerance, and overall suitability for AAE (Sample 2) (medians, 25th and 75th percentiles, adjacent values, and outliers). Lack of statisticaly significant differences (*p* = 0.721).

**Table 1 animals-14-01149-t001:** Definitions of dog behaviors according to ChatGPT.

Question	Answer Given by ChatGPT	Note
How does an aggressive dog behave?	The dog engages in actions aimed at causing harm or intimidating people or objects. It displays threatening signals, may impulsively turn around, attempt to bite, show teeth, growl, bark, tug at objects, or kick.	1
How does an anxious dog behave?	The dog tries to escape (by seeking shelter, hiding in a corner/behind the owner, attempting to go outside). Muscle trembling, turning the head away, tucking the tail between the hind legs, licking, excessive panting, and even urination may be observed.	2
How does a grounded dog behave?	The dog is calm and relaxed. Positive interactions with people are evident. The dog exhibits emotions that are appropriate to the situation; it is capable of showing joy, curiosity, and interest. If given food, the dog has a good appetite.	3
How does an excited dog behave?	The dog is excessively active, and the caretaker could find it difficult to calm the dog down. The animal exhibits behaviors such as running around, jumping, making sudden movements, or displaying overall hyperactivity. The dog may jump on people, furniture, or other objects in its surroundings. It might have difficulty concentrating on tasks or commands and responds only when the caretaker repeats the command several times.	4
How does a reactive dog behave?	The dog is in constant motion, running, jumping, spinning around, and displaying high levels of physical activity, even in situations where most dogs would be calmer. Its need for activity seems insatiable. The dog cannot focus on a single task or command, and it may produce various sounds (whining, barking, growling). The dog may also engage in destructive behaviors.	5

**Table 2 animals-14-01149-t002:** Inter-rater reliability in Subtests 1–3.

Subtest 1	B	R	O	V
B	-			
R	0.350	-		
O	0.194	0.277	-	
V	0.665	0.352	0.000	-
T	0.665	0.705	0.263	0.699
Subtest 2				
B	-			
R	0.267	-		
O	0.468	0.213	-	
V	0.611	0.000	0.280	-
T	0.684	0.382	0.586	0.613
Subtest 3				
B	-			
R	0.711	-		
O	0.403	0.600	-	
V	0.539	0.542	0.696	-
T	0.522	0.741	0.549	0.350

**Table 3 animals-14-01149-t003:** Inter-rater reliability in Subtests 4–6.

Subtest 4	B	R	O	V
B	-			
R	0.709	-		
O	0.619	0.761	-	
V	0.354	0.316	0.404	-
T	0.623	0.549	0.605	0.727
Subtest 5				
B	-			
R	0.347	-		
O	0.182	0.827	-	
V	0.543	0.000	0.101	-
T	0.441	0.518	0.248	0.000
Subtest 6				
B	-			
R	0.784	-		
O	0.579	0.872	-	
V	0.279	0.673	0.595	-
T	0.546	0.821	0.885	0.516

**Table 4 animals-14-01149-t004:** Inter-rater reliability in Subtests 7–9.

Subtest 7	B	R	O	V
B	-			
R	0.363	-		
O	0.472	0.689	-	
V	0.540	0.692	0.597	-
T	0.728	0.478	0.766	0.656
Subtest 8				
B	-			
R	0.351	-		
O	0.474	0.893	-	
V	0.687	0.596	0.753	-
T	0.435	0.640	0.804	0.771
Subtest 9				
B	-			
R	0.511	-		
O	0.508	0.805	-	
V	0.601	0.438	0.663	-
T	0.486	0.198	0.291	0.747

**Table 5 animals-14-01149-t005:** Inter-rater reliability in assessments of sociability, touch tolerance, and overall suitability for AAE.

Sociability	B	R	O	V
B	-			
R	0.288	-		
O	0.597	0.646	-	
V	0.502	0.149	0.214	-
T	0.701	0.368	0.791	0.432
Touch tolerance				
B	-			
R	0.469	-		
O	0.697	0.717	-	
V	0.290	0.590	0.545	-
T	0.683	0.435	0.799	0.628
Overall rating				
B	-			
R	0.665	-		
O	0.386	0.835	-	
V	0.594	0.531	0.427	-
T	0.406	0.717	0.807	0.486

## Data Availability

The data presented in this study are available upon request from the corresponding author.

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
