# Peer review of "A Method for Assessing Dogs in a Test Evaluating Dogs’ Suitability for Animal-Assisted Education"

_animals, 2024, doi:10.3390/ani14081149_

Round 1

Reviewer 1 Report

Comments and Suggestions for Authors

Wondering about CAE terminology? This term will not likely come up in most key word searches. Historically Animal Assisted Education has been used in the ‘industry’.  Also, there is an international movement for uniform terminology updates. The article was recently published and has been agreed upon internationally be several organizations. Might be worth consideration. It appears your definitions are based on a single American Organization called Pet Partners. It may not matter much for this particular paper,  but I think it worth the mention as this journal has international readership and Pet Partners may not have many international teams and may not be well known in other countries. It may be better to follow guidelines in the following paper, and look to international organizations such as Animal Assisted Intervention International or International Association for Human-Animal Interaction Organizations as they both have officially adopted this terminology (rather than the American Pet Partners which is not necessarily reflective of an international presence).

Binder, A. J., Parish-Plass, N., Kirby, M., Winkle, M., Skwerer, D. P., Ackerman, L., ... & Wijnen, B. (2024). Recommendations for uniform terminology in animal-assisted services (AAS). Human-Animal Interactions12(1).

On page 1, you describe AAT in volving ‘patients’. The term ‘patient’ typically indicates hospital programs- AAT is also included in private practices, in school based services (that are outside the scope of educational). You may be more reader friendly and accurate using the term clients, so it is not limited to only medical model.

 While Poland does not have any legal regulations, I wonder if you could include some of the existing evaluations such as PADA?

Line 96: Could better define the qualifications of the ‘experienced trainer’, the veterinarian, the handler, etc. Were they specialists, formally trained, etc.?

Beginning at line 99, there is a description of a suitability test [for CAE], however there is no indication for the reader as to what each item is assessing, no definition of suitability. How do each of these items indicate suitability?  If I am not mistaken, much of this article is based on these items- were they also evaluated to be reliable and valid at some point (even if in history?) I am not convinced readers will make the leap for being asked to believe these are indicators of suitability …leading to AI also finding the same outcomes. How do we know there are indicators of suitability?

When using ChatGPT definitions, does it produce references for the definitions it came up with?  Were these generated definitions/descriptions evaluated by actual behaviorists for the purposes of this study, or perhaps are the authors qualified behaviorists ? I see some differences between the generated ChatGPT descriptions and traditional veterinary/behavior texts definitions.

 Starting at line 225: You indicate that the standard procedure is that there is a dedicated training course, pass final exam, re-evaluate every year….. but this is specifically for Poland. In the US, there is not a process for healthcare and human service providers to be evaluated. Many people go through organizations such as Pet Partners- however, their evaluation is specifically for VOLUNTEER VISITORS and it specifically states in the manuals that the evaluation is not meant for AAT/AAE with professionals. They also only re-evaluate every two years. So, it may be wise to indicate that your example is specific to that book, and not representative of a sampling of organizations that evaluate ‘therapy dogs’.  That term is also problematic as it does not allow for differences in AAA versus skilled services of AAT/AAE.

Line 232: Any dog is capable of momentary aggressive behavioral outcomes especially if they are in fear, it is not necessarily an indication that they are always an aggressive dog and inappropriate.

There was also information missing such as how novelty can create anxiety or fear, there may be differences in a dog who is evaluated in a novel setting versus a familiar setting such as where it will be working.

Comments on the Quality of English Language

General grammar, punctuation- there are several misplaced hyphens throughout this paper.

Author Response

Dear Reviewer,

We would like to thank you for taking the time to review our manuscript. We respond to your recommendations below.

- Reviewer wrote: Wondering about CAE terminology? This term will not likely come up in most key word searches. Historically Animal Assisted Education has been used in the ‘industry’.  Also, there is an international movement for uniform terminology updates. The article was recently published and has been agreed upon internationally be several organizations. Might be worth consideration. It appears your definitions are based on a single American Organization called Pet Partners. It may not matter much for this particular paper,  but I think it worth the mention as this journal has international readership and Pet Partners may not have many international teams and may not be well known in other countries. It may be better to follow guidelines in the following paper, and look to international organizations such as Animal Assisted Intervention International or International Association for Human-Animal Interaction Organizations as they both have officially adopted this terminology (rather than the American Pet Partners which is not necessarily reflective of an international presence).

- Our response: Thank you for your comment. We have changed the date of CAE to AAE where required. In some places, however, we have retained the term canine-assisted education to indicate that we are referring to this species, which has its own specificity. However, given your first general comment regarding terminology, we have removed the abbreviation CAE from the manuscript. We left the reference to Pet Partners in the text because many authors have referred to it over the years, probably because this organization is a continuation of the Delta Society.  At the same time, as you suggested, we added information about the international movement for uniform terminology and provided newer proposed terminology according to Binder et al. (2024) (lines 43-230). In the future, we will likely only use the latter terminology.

- Reviewer wrote: On page 1, you describe AAT in volving ‘patients’. The term ‘patient’ typically indicates hospital programs- AAT is also included in private practices, in school based services (that are outside the scope of educational). You may be more reader friendly and accurate using the term clients, so it is not limited to only medical model.

- Our response: Thank you for this advice. We replaced the word patient with the word client (line 231).

- Reviewer wrote: While Poland does not have any legal regulations, I wonder if you could include some of the existing evaluations such as PADA?

- Our response: Our test was used by one of the organizations before the PADA test was released. It is very likely that in the future we will want to verify our proposal for assessing dogs using the PADA test. As you point out, PADA is a well-structured test and would probably be worth using widely, so we quote it in the new version of our manuscript (lines 244-245).

- Reviewer wrote: Line 96: Could better define the qualifications of the ‘experienced trainer’, the veterinarian, the handler, etc. Were they specialists, formally trained, etc.?

- Our response: We improved description of the competences of these people (lines 526-528). The exception was an ordinary dog owner who did not have any qualifications related to dogs, so we did not add any description to him.

- Reviewer wrote: Beginning at line 99, there is a description of a suitability test [for CAE], however there is no indication for the reader as to what each item is assessing, no definition of suitability. How do each of these items indicate suitability?  If I am not mistaken, much of this article is based on these items- were they also evaluated to be reliable and valid at some point (even if in history?) I am not convinced readers will make the leap for being asked to believe these are indicators of suitability …leading to AI also finding the same outcomes. How do we know there are indicators of suitability?

- Our response: Thank you for this important comment. In new version of manuscript we added information about the purpose of each stage (lines 334-362). When it comes to evaluating whether this test was useful, we have information that it helped eliminate dogs that were too fearful and/or aggressive. Because the dog owners know each other, few years later information reached that these were the right decisions. This is in line with our belief, and we write about it in Introduction, that the initial test of the suitability of dogs for therapeutic activities is mainly to look for dogs that are not suitable for this purpose. To our knowledge, there has been no other method to evaluate this test.

- Reviewer wrote: When using ChatGPT definitions, does it produce references for the definitions it came up with?  Were these generated definitions/descriptions evaluated by actual behaviorists for the purposes of this study, or perhaps are the authors qualified behaviorists? I see some differences between the generated ChatGPT descriptions and traditional veterinary/behavior texts definitions.

- Our response: ChatGPT did not provide references to its definitions. Interestingly, when we ask for references, we get a list of references, but no definitions. Both authors of the manuscript are COAPE certified behaviorists (we added this information in new version of manuscript, line 365). Additionally, the corresponding author deals with dog behavior scientifically and lectures in this field at the university. Of course, we noticed some differences between the definitions proposed by AI and those found in professional studies. We added this information in the new version of the manuscript (lines 705-707).

- Reviewer wrote: Starting at line 225: You indicate that the standard procedure is that there is a dedicated training course, pass final exam, re-evaluate every year….. but this is specifically for Poland. In the US, there is not a process for healthcare and human service providers to be evaluated. Many people go through organizations such as Pet Partners- however, their evaluation is specifically for VOLUNTEER VISITORS and it specifically states in the manuals that the evaluation is not meant for AAT/AAE with professionals. They also only re-evaluate every two years. So, it may be wise to indicate that your example is specific to that book, and not representative of a sampling of organizations that evaluate ‘therapy dogs’.  That term is also problematic as it does not allow for differences in AAA versus skilled services of AAT/AAE.

- Our response: Thank you for this suggestion. In the new version of the manuscript, we started the paragraph with the assumption that we were writing about many organizations in Poland (so not all of them), according to the referred book (lines 640-641 and 643). In Poland, exams are usually taken for AAI or, less often, separately for AAT and AAE. There are good reasons for both approaches, but we did not want to expand our manuscript too much, especially since it was not our main topic. Nevertheless, you raised a very interesting topic in your comment.

- Reviewer wrote: Line 232: Any dog is capable of momentary aggressive behavioral outcomes especially if they are in fear, it is not necessarily an indication that they are always an aggressive dog and inappropriate. There was also information missing such as how novelty can create anxiety or fear, there may be differences in a dog who is evaluated in a novel setting versus a familiar setting such as where it will be working.

- Our response: Thank you for drawing attention to this aspect. In the previous version of the manuscript we incorrectly presented the relationship between novelty and aggression, so we improved it (lines 648-650). We also added the sentence that it should also be taken into account that every dog may behave aggressively in certain circumstances, especially in fear. Therefore, the manifestation of aggression does not always mean that the dog is generally aggressive and should be excluded from AAI (lines 698-700).

The revised manuscript has been spell-checked and grammar-checked by a professional translator and a native English speaker to eliminate stylistic inconsistencies, and to improve overall readability and clarity of presentation.

We are grateful for all the valuable comments which were helpful in improving the manuscript. We hope our revision and response would be satisfactory. If needed, we can further improve the manuscript.

Reviewer 2 Report

Comments and Suggestions for Authors

Overall:  Overall good, but the introduction, methods, and discussion need additional information. 

Abstract:  Overall the abstract and simple summary do a good job of summarizing the paper and providing adequate details regarding the suitability of these tests for assessing canines for educational work.  To further clarify what was done in this study, it would be helpful if the authors mentioned that traditional BR methods use behavioral definitions that can differ between raters/observers so the use of AI definitions can help standardize the measures and that this study aimed to look specifically at the role of AI in BR.  It might also be helpful to add that behavioral definitions provided by AI were based on emotional states and that raters included professional behaviorists, veterinarians, and the dog owners. 

Line 31:  The term “SR” needs to be defined

Introduction: Overall the introduction does a good job of introducing the topic of animal-assisted activities and how animals are involved in these programs.  With regards to behavioral assessments, the authors bring up BR and SR.  It would be helpful if they also mentioned Qualitative Behavioral Assessments (QBA) as well since this is very commonly used to look at the ability of different animals across species as it pertains to their ability to be involved in AAA. 

It would also be helpful in the introduction to include how BR is usually done and why AI was included in this specific study.  What is wrong with traditional BR that AI needed to be tested at all?  OR how can AI improve current methods?  These are important questions that should be addressed in the introduction to justify the study.  Would it help the broader public to assess dog behavior more accurately?  Why is this important?

Materials and Methods:  Overall, this section was thorough and provided adequate detail regarding the study and design of the experiment.   The authors should include information regarding how they determined the questions they asked ChatGPT regarding behavior and why they focused on certain words like “aggression”.  What led up to the formation of those specific questions and why didn’t they formulate questions such as “what are behavioral indicators of high stress that might lead to a dog biting?”  This is important to know since various questions in ChatGPT can get differing results. 

Results:  Results were clear and presented in a format that was thorough and easy to understand. 

Discussion:  Overall, the discussion is good.  Additional limitations need to be addressed.  Most importantly, the authors need to address the issue that the definitions provided by AI were not validated by dog behaviorists (even ChatGPT says that information needs to be checked and validated).  If the behaviors suggested by ChatGPT were compared against literature that validated the behavior descriptions, this needs to be included in the methods section.   

The limitations section of the discussion should also include any potential variations of definitions that come from CHatGPT and other AI programs as potentially problematic since ChatGPT often creates very colorful descriptions based on creative text it finds online.  There should be a lot of caution around using ChatGPT blindly so it needs to be included in the discussion.  Mentioning the potential for AI or the development of a more validated tool with the help of AI could be emphasized, but the limitations section should be more extensive as a result. 

Author Response

Dear Reviewer,

We would like to thank you for taking the time to review our manuscript. We respond to your recommendations below.

- Reviewer wrote: Abstract:  Overall the abstract and simple summary do a good job of summarizing the paper and providing adequate details regarding the suitability of these tests for assessing canines for educational work.  To further clarify what was done in this study, it would be helpful if the authors mentioned that traditional BR methods use behavioral definitions that can differ between raters/observers so the use of AI definitions can help standardize the measures and that this study aimed to look specifically at the role of AI in BR.  It might also be helpful to add that behavioral definitions provided by AI were based on emotional states and that raters included professional behaviorists, veterinarians, and the dog owners. Line 31:  The term “SR” needs to be defined.

- Our response: Thank you for this advice. We improved Simple Summary and Abstract. Please note, however, that we are limited by the word limit and content guidelines for these summaries. We also defined the term SR.

- Reviewer wrote: Introduction: Overall the introduction does a good job of introducing the topic of animal-assisted activities and how animals are involved in these programs.  With regards to behavioral assessments, the authors bring up BR and SR.  It would be helpful if they also mentioned Qualitative Behavioral Assessments (QBA) as well since this is very commonly used to look at the ability of different animals across species as it pertains to their ability to be involved in AAA.

- Our response: We have added information about this method in the new version of the manuscript (lines 263-265).

- Reviewer wrote: It would also be helpful in the introduction to include how BR is usually done and why AI was included in this specific study. What is wrong with traditional BR that AI needed to be tested at all? OR how can AI improve current methods? These are important questions that should be addressed in the introduction to justify the study. Would it help the broader public to assess dog behavior more accurately?  Why is this important?

- Our response: In the new version of the manuscript, we have included information on how the BR method is implemented (lines 257-260). The second issue, regarding the disadvantages of differences in the interpretation of behavior by observers in the BR method, was already described in the first version of the manuscript, at the end of the next paragraph (lines 276-279). We also pointed out that AI could help standardize definitions of behavior. Following your suggestion, we also added information on how AI can help assess and predict dog behavior and prevent problems before they occur (lines 280-281).

We could have used our own definitions instead of the definitions proposed by AI, but we wanted to point out that there is a possibility of using AI. Additionally, AI is constantly being developed and we are convinced that its role in scientific work will grow. On the other hand, we find this a bit disturbing, but this topic would go well beyond the purpose of this manuscript. Regarding your further comment, we tried to answer it in the last paragraph of Discussion (we have expanded it in the revised version of the manuscript). In short, the method we have described is not yet perfect, but it may pave a new path in assessing the suitability of dogs for various tasks.

- Reviewer wrote: Materials and Methods:  Overall, this section was thorough and provided adequate detail regarding the study and design of the experiment. The authors should include information regarding how they determined the questions they asked ChatGPT regarding behavior and why they focused on certain words like “aggression”.  What led up to the formation of those specific questions and why didn’t they formulate questions such as “what are behavioral indicators of high stress that might lead to a dog biting?” This is important to know since various questions in ChatGPT can get differing results.

- Our response: Thank you for this comment. First, we decided that we wanted the behavior descriptions for each stage to describe the dogs' emotions or emotional states. To our knowledge, no one has done this before to create a BR assessment method. We then transformed these emotional states into behaviors using AI. The emotions/emotional states we used were quite obvious when we watched the test videos. We agreed that one of the dogs was definitely aggressive, some were anxious, some were grounded, and some were excited and some were reactive. This division resulted from our previous experience with dogs and these observations. We focused on words like aggressive or anxiety because, as we wrote in the Introduction, dogs that are aggressive and excessively anxious are not suited for AAI (lines 239-240). We decided that it would be best to ask the AI the simplest questions, because we wanted to get substantial answers. However, we tried to ask other questions, but received evasive answers. To the question "what are behavioral indicators of high stress that might lead to a dog biting?" the AI only gives you signs of high stress that might lead to biting. Of course, these signs are not yet aggression, and some dogs choose the aggression (Fight) strategy, other dogs choose the other 4F strategy. It was such speculations that led us to such simple questions asked to AI. However, it is difficult to describe it in the manuscript, we believe that everyone should see for themselves what the possibilities and limitations of using AI are.

- Reviewer wrote: Results:  Results were clear and presented in a format that was thorough and easy to understand.

- Our response: Thank you very much for this approbation!

- Reviewer wrote: Discussion: Overall, the discussion is good. Additional limitations need to be addressed.  Most importantly, the authors need to address the issue that the definitions provided by AI were not validated by dog behaviorists (even ChatGPT says that information needs to be checked and validated).  If the behaviors suggested by ChatGPT were compared against literature that validated the behavior descriptions, this needs to be included in the methods section.

- Our response: We apologize for not writing this before, but both authors of the manuscript are canine behavior experts (COAPE certified animal behaviorists). We have updated this information in the new version (line 365). Additionally, the corresponding author deals with dog behavior scientifically and lectures in this field at the university.

- Reviewer wrote: The limitations section of the discussion should also include any potential variations of definitions that come from CHatGPT and other AI programs as potentially problematic since ChatGPT often creates very colorful descriptions based on creative text it finds online.  There should be a lot of caution around using ChatGPT blindly so it needs to be included in the discussion.  Mentioning the potential for AI or the development of a more validated tool with the help of AI could be emphasized, but the limitations section should be more extensive as a result.

- Our response: We agree with you about limiting the use of AI. We have expanded the limits and included the use of AI (lines 705-711).

We are grateful for all the valuable comments which were helpful in improving the manuscript. We hope our revision and response would be satisfactory. If needed, we can further improve the manuscript.

Reviewer 3 Report

Comments and Suggestions for Authors

Authors used definition of behavior based on AI, which may be an interesting approach.  I feel there is a flaw in the experiment design for two reasons.  One is that since the raters are familiar with the dogs, they may not have relied on the definition provided by AI.  Possible interview with the raters may suggest how much these definition may have helped the.  The second point is that rather than comparing two different methods of evaluation, they have used control raters with same background, asking them to rate the behavior without the definition provided by AI.  

Author Response

Dear Reviewer,

We would like to thank you for taking the time to review our manuscript. We respond to your comments below.

- Reviewer wrote: Authors used definition of behavior based on AI, which may be an interesting approach.  I feel there is a flaw in the experiment design for two reasons.  One is that since the raters are familiar with the dogs, they may not have relied on the definition provided by AI.  Possible interview with the raters may suggest how much these definition may have helped the.  The second point is that rather than comparing two different methods of evaluation, they have used control raters with same background, asking them to rate the behavior without the definition provided by AI.

- Our response: Thank you for your observations. We believe that on the one hand you are right, but on the other hand, our approach was also correct. Let us explain why we took this approach in the methodology. You are right that raters are familiar with the dogs. Of course, no one should entrust the assessment of dogs' behavior to people who do not know anything about dogs, therefore, dog behavior should always be assessed by specialist. Since they know something about dogs, their assessment (even with AI guidance) will always be subjective, burdened by their previous experiences. On the other hand, if raters had no idea about dogs, their assessment would also be biased, because we believe everyone has some idea about dogs, wrong concept in such case. We also ask you to note that, as we wrote in lines 528-529, the raters used Table 1 excluding column 1 containing questions, which means that they did not know that the given description of behavior concerns the definition of some emotional state. In other words, the information that the behavior described in the table defines the behavior of an aggressive dog (rated 1), and that the behavior described in the table defines the behavior of an anxious dog (rated 2) was known only to us. Following your comment, we agree with you that in the future it would be worth asking raters how helpful the AI assistance was, and perhaps good idea would be to compare their ratings without and with AI assistance. In this case, however, we consider this manuscript to be a tentative one; it is a preliminary proposal of a method which, as we try to explain in the last chapter of Discussion, should be further verified and improved. We would like to assure you that we want to develop this method in the future, including checking the direction you indicated. We would also like to mention that thanks to your comment, we have added a sentence to the Discussion regarding doubts about the use of AI by human raters (lines 705-713).

Regarding your second objection, we would like to point out that it would be entirely right if it were about the ratings themselves. However, we did not compare the ratings, but the distributions of inter-rater reliabilities between observers, because our goal was to find out whether the proposed evaluation method would produce good consistency between raters, comparable to other method. We think your point makes sense, but please accept this explanation, which we think is sensible, too.

Thank you for your comments, which will certainly help us in further planning. We believe that this is an important topic that should be explored in more detail in the next full-length publication. This is a broad topic, so we are not able to cover it in the current manuscript. However, we will continue to work on this topic and invite you to contact us and cooperate on it.

Again, we would like to thank the reviewer for reading our manuscript and reviewing it. We hope you will be satisfied with the improvements and the explanations we have made.

Reviewer 4 Report

Comments and Suggestions for Authors

The research was well conducted and the topic is novel and interesting 

It could beba good idea to compare definitions gave by chatGPT with other definitions commonly used in the field

Author Response

Dear Reviewer,

We would like to thank you for taking the time to review our manuscript. We respond to your comments below.

- Reviewer wrote: The research was well conducted and the topic is novel and interesting. It could be a good idea to compare definitions gave by chatGPT with other definitions commonly used in the field.

- Our response: We would like to thank you for this approval. In response to your suggestion, we would like to confirm that we have compared the definitions proposed by AI with scientific ones from several sources. There were some minor differences in definitions. For example, the definition of aggressive behavior given here is: ,,The dog engages in actions aimed at causing harm or intimidating people or objects. It displays threatening signals, may impulsively turn around, attempt to bite, show teeth, growl, bark, tug at objects, or kick”, whereas definition given by Farhoody et al. (2018, doi: 10.3389/fvets.2018.00018) is: ,,Typical signs of moderate aggression in dogs include barking, growling, and baring teeth. More serious aggression generally includes snapping, lunging, biting, or attempting to bite”. In the revised manuscript, we added an appropriate note in the discussion on this topic (lines 705-711). We also added information, that both authors of the manuscript are COAPE certified animal behaviorists (line 365). Additionally, the corresponding author deals with dog behavior scientifically and lectures in this field at the university. Therefore, we felt competent to be able to ask the AI the right questions and to be able to assess the accuracy of its answers.

Again, we would like to thank the reviewer for reading our manuscript and reviewing it. We hope you will be satisfied with the explanations we have made.

Round 2

Reviewer 3 Report

Comments and Suggestions for Authors

Thank you for replying to my comments.  I hope your attempt would lead to an even better behavior ratings in the future.